# Numerical Modeling of the Thermal Insulating Properties of Space Suits

**DOI:** 10.3390/ma17030648

**Published:** 2024-01-29

**Authors:** Michał Dzięgielewski, Ryszard Korycki, Halina Szafrańska, Marcin Barburski

**Affiliations:** 1Faculty of Material Technologies and Textile Design, Lodz University of Technology, Zeromskiego, 116, 90-924 Lodz, Poland; 202706@edu.p.lodz.pl; 2Department of Mechanical Engineering, Informatics and Chemistry of Polymer Materials, Lodz University of Technology, Zeromskiego, 116, 90-924 Lodz, Poland; 3Department of New Technologies and Materials, Radom University, Chrobrego Str., 27, 26-600 Radom, Poland; h.szafranska@uthrad.pl; 4Institute of Architecture of Textiles, Lodz University of Technology, Zeromskiego, 116, 90-924 Lodz, Poland; marcin.barburski@p.lodz.pl

**Keywords:** numerical modeling, thermal insulating properties, space suit

## Abstract

The purpose of this study was to model the thermal insulating properties in an exemplary multi-layer layup of space suits utilizing computer simulation techniques and physics and mathematical models. The main system responsible for thermal insulation is the Thermal Micrometeoroid Garment (TMG) material layup. Its structure consists of eight layers with different functions. The utilized textile materials are Rip-Stop-type fabrics, strengthened with the addition of a stronger fiber at fixed intervals. The state variable in thermal problems is the temperature field inside the analyzed TMG. The results obtained from the computer simulation were compared to verification calculations from the mathematical model, which allowed for an assessment of the models’ quality and the obtained results. Two numerical models were analyzed in Ansys Workbench software. This enabled an assessment of the model’s quality and the possible discrepancies. The modeling of the structure was carried out using the Finite Element Method. The possibility of using this exemplary material layup for a suit was verified using normalized data for an adult in outer space.

## 1. Introduction

### 1.1. Outer Space Conditions

The specific conditions of outer space are defined by a variety of factors, which are difficult to protect oneself from.

In the 10^−4^–10^−15^ Pa space vacuum, some materials have an outgassing tendency, which entails a release of volatile compounds. The accumulation of unwanted substances on cooled surfaces causes a reduction in the efficacy of sensors utilized in outer space [1]. The materials used therein undergo tests to determine TML and CVCM, which, according to NASA guidelines, should have a TML < 1% and CVCM < 0.1% [2].

The threat to human health resulting from exposure to the space vacuum primarily involves air expansion in the lungs, leading to death. Another issue is the reduction in the blood boiling point in low-pressure conditions (ebullism). It poses a risk of cutting off oxygen delivery to the brain and of the creation of gas pockets that can block blood vessels [3]. Yet another danger is hypoxia caused by the lack of elemental oxygen in a vacuum. A person will lose consciousness after several seconds [4].

Atomic oxygen is formed as a result of the reaction of elemental oxygen contained in the upper layers of the earth’s atmosphere with short-wave ultraviolet (UV) radiation. This form of oxygen strongly reacts with materials containing carbon, nitrogen, sulfur and hydrogen bonds and with alloys based on silver, copper and osmium. Material degradation occurs, and the layers protecting from atomic oxygen may become defective in places with cuts and notches [1].

UV radiation primarily comes from the sun, causing uncontrollable polymer material crosslinking, damaging them and impairing their functionality [1]. In a high vacuum, this results in deep surface discoloration caused by insufficient oxygen levels. 

The detrimental effect of UV radiation on humans mainly entails skin damage such as burns and tumors. Ultraviolet radiation damages the retina and is one of the main causes of cataracts [5]. Apart from UV radiation, there is also ionizing radiation. Its intensity, to a large extent, depends on the activity of the sun. This type of radiation leads to irreversible changes at the molecular and cellular levels [1].

Due to the vacuum present in outer space, convection and conduction have negligible impacts on all thermal phenomena. Only the heat from the sun’s radiation, reflected by the Earth or other celestial bodies in space, affects the perceived temperature. It is assumed that temperature fluctuations due to the position relative to the Sun are between −160 °C and 130 °C [6]. The intensity of objects’ thermal changes depends on their position in relation to the Sun and other bodies, on the thermal–optical properties of the considered object and on the radiation exposure time [1]. For materials covered with a protective coating, knowing their CTE is imperative. If there is a discrepancy between the CTEs of the coating and core materials, the layers exposed to cyclic temperature swings are susceptible to cracking and peeling [1].

Micrometeoroids constitute a significant danger due to their high velocities whilst traveling through space. Their average velocity is about 10 km/h, but a velocity as high as about 60 km/h has also been recorded [3]. Presently, it is not possible to avoid smaller micrometeoroids, whose impact is comparable to a projectile fired from a firearm. 

### 1.2. Spacesuits

There are two main types of space suits. The first one is utilized only in the rocket launch phase and when re-entering the earth’s atmosphere. Its function is protection from high temperatures and from pressure changes due to entering the atmosphere. It is also a safeguard when landing on the Earth [7].

The second type of space suit is used by the International Space Station crew (Extravehicular Mobility Unit—EMU) for extravehicular activities (EVAs), such as the maintenance of objects located in space [8]. It is designed to allow the crew to survive for 7 h while working in normal conditions and for 6 h while working in conditions with the highest possible exposure to the sun’s radiation. As a multifunctional system, the suit fulfills the following requirements [8]: Protects against the space environment;Enables mobility in outer space conditions;Contains a life support system;Maintains communication between the crew and the ISS base station.

The main system responsible for thermal insulation is the Thermal Micrometeoroid Garment (TMG) material layup. Its structure consists of eight layers with the following functions, starting from the human body [9,10,11,12] (Figure 1). 

Layer 1: Human skin.Layer 2: LCVG (Liquid Cooling and Ventilation Garment): a two-layered layup with a water-fed tube system. The flow of liquid facilitates the cooling of the body in extreme conditions. The first nylon-made layer ensures good adherence to the body. The second one made of spandex or nylon has polymer tubes.Layer 3: The Pressure Garment Layer contains the oxygen needed for breathing. Furthermore, the gas within this layer allows the astronauts to maintain normal blood pressure. It is a layer of nylon covered with thermoplastic urethane.Layer 4: The Dacron layer provides the stiffness of the gas-filled layer.Layer 5: The neoprene-covered nylon prevents the tearing of the material layers.Layer 6: Seven layers of aluminum-covered mylar create the main thermo-insulating layer. Mylar is the foil formed as a result of PET polyester stretching. It exhibits excellent thermal properties: the melting point is 254 °C, and in the temperature range from 50 to 200 °C, it maintains thermal stability. The aluminum coverage provides a safeguard from heat radiation between the layers. The distances between the layers are negligible. To simplify the calculations, the seven layers of aluminum-covered mylar are shown as one layer.Layers 7, 8, and 9: The three-layer external structure (Ortho-fabric) ensures protection from accelerated micrometeoroids (Kevlar) and fire (Goretex) and water (Nomex) resistance.

The utilized textile materials are Rip-Stop-type fabrics, enhanced with the addition of a stronger fiber at fixed intervals. They are light, durable and impermeable to air and water vapor [13].

The available literature does not provide data on heat conduction and heat insulation in a space suit. The thermal insulation problem chiefly involves textile materials, which are the main components of this gear. The best reference regarding the global approach to thermal comfort is Li [14]. The author analyzes the neurophysiological and physiological approach to thermal comfort, physical mechanisms of temperature and moisture sensations, as well as dynamic heat and moisture transfer in textile fabrics. The problem is determined using differential equations accompanied by a set of boundary and initial conditions. A dynamic model of liquid water transfer coupled with moisture sorption, condensation and heat transfer in porous textiles can be developed by incorporating the physical mechanism of liquid diffusion in porous textiles into a coupled heat and moisture transfer model, cf., Li, Zhu [15]. The shape optimization problem for transient heat conduction was solved within the context of isogeometric analysis by Wang et al. [16]. An adjoint sensitivity analysis, which accounts for possible discontinuities in the objective functionality, was performed analytically and subsequently discretized. In terms of fire exposure and radiation, it is, in some ways, analogous to a firefighter uniform [17,18,19].

The purpose of this article is to model the thermal insulating properties in an exemplary multi-layer layup of a space suit, both its entirety as well as the specific layers, utilizing computer simulation techniques and physics and mathematical models. The results obtained in the computer simulation were compared to the results from the mathematical model, which allowed for a quality assessment of the models. The possibility of using an exemplary material layup of the suit was verified using normalized data for an adult in outer space. 

This article contains the following new insights: A comparison of the results obtained using a computer simulation with the verification calculations for the simplified model of heat exchange. Two numerical models were analyzed: the verification model under simplified boundary conditions to assess the results’ credibility and the actual model with parameters similar to outer space conditions. This enabled an assessment of the model’s quality and of the possible discrepancies.An analysis of the thermo insulating properties of a space suit for a person in extreme outer space conditions. The specific conditions of outer space were defined by a variety of factors (i.e., the space vacuum, atomic oxygen, UV-radiation, temperature, micrometeoroids).The discussion on possible uses of the proposed model solution.

## 2. Materials and Methods

### 2.1. Metabolism and Heat Exchange

A part of the energy generated by the human body is devoted to its functioning; the rest is released to the environment. The intensity of these changes depends on [20]: human metabolism, thermal insulation of clothing and the microclimate. 

Human metabolism determines several processes occurring within an organism as a result of chemical reactions and is a source of heat in a human body [20].
(1)q=M(1−η)±P±V±K±R±C

Balanced heat flux density can be converted into heat flux with a known surface area of the human skin. There are many formulas describing the relationship between surface area, body weight and human height—an example is the Haycock Formula.
(2)$$S=0.024265\cdot L^{0.3964}\cdot M_{b}^{0.5378}$$

The metabolic rate is most commonly linked to physical activity [20], and it can be determined based on a set norm [21]. The efficiency coefficient η = (17–25)%, which gives on average η = 20%. The perceived thermal comfort is determined by CT = (37 ± 0.3) °C and a skin temperature of about 33 °C [20]. During extravehicular activity, it is assumed that for 7 h of work, the average metabolic rate does not exceed 300 W/m^2^, and for 1 h of work, it does not exceed 470 W/m^2^. Work under extreme conditions is permissible for 15 min, with the maximal rate of 600 W/m^2^ [8]. In laboratory research, the highest metabolic rate recorded was 880 W/m^2^, while during the Apollo mission to the Moon, the lowest rate was at 144 W/m^2^ [22].

Thermal conductivity is a process of energy exchange between particles not undergoing macroscopic transformations. The base description is given by Fourier’s law. Space suits are constructions comprising multiple curves, described at any point by a variable radius. Therefore, conduction is described for a n-layered construction with variable curves in relation to the length unit [23].
(3)$$Q=\frac{2\pi \left(T_{w1}-T_{wn}\right)}{\sum_{i=1}^{n}\left(\frac{1}{\lambda_{i}}\cdot ln\frac{r_{i+1}}{r_{i}}\right)}$$

Heat radiation occurs from the suit’s surface (gray body) to the environment. The Stefan–Boltzmann law is described by the following equation [20]: (4)$$Q=A\cdot \sigma \cdot \varepsilon \cdot T^{4}$$

The aim of the simulation is the assessment of the thermal insulation of the space suit during prolonged exposure to the extreme conditions in outer space. In this case, the steady-state conditions might be adapted for simplification. The equation of the thermal conductivity state is as follows [24]: (5)$$\lambda \nabla T+\varepsilon F\sigma \left({T_{s}}^{4}-{T_{a}}^{4}\right)+h(T_{s}-T_{a})=0$$

The state variable in thermal conductivity issues is the temperature [20,25,26]. It creates a state area since it is possible to be determined at every point of the suit. 

### 2.2. Numerical Modeling

Two numerical models were analyzed in Ansys Workbench software (Ansys Workbench v. 2023)—the verification model under the simplified boundary conditions to assess the results’ credibility and the actual model with parameters similar to the outer space conditions. This enabled an assessment of the model’s quality and of the possible discrepancies.

The modeling of the structure was carried out with the Finite Element Method (FEM), and this process entailed the following steps [27]: Approximation—the application of the material data and geometrical conditions to ensure similarity between the model and the actual object.Discretization—the transformation of a continuous mathematical model into a discrete model, divided into a finite number of elements.The solution—the determination of the temperature area and the heat flux density.Verification—the assessment of the quality of the obtained results, which will allow for the applied modifications.

The geometry of a finite element influences the quality of the obtained solution [28,29]. For the issues with directional changes, quadrilateral and hexagonal elements are proven to be more efficient, as their proportionality coefficients are notably larger than those of triangular elements. These help to significantly decrease the number of elements in the model, which also lessens the computational requirements for the computer [30]. However, taking into account the sensitivity of the quadrilateral elements to the model’s geometry, it is impossible to obtain satisfactory results for more complex shapes.

There are no additional sources of heat in the adopted models. The mean value of the thermal conductivity coefficient is also assumed for the extreme temperatures. The ideal interaction between the layers‘ surfaces was introduced, which facilitates the examination of heat conduction only. The models, in addition to the material layers of the suit, also encompass the outer layers of the human skin. Thus, the aforementioned boundary condition is CT. Human skin temperature may differ due to environmental conditions and the heat flux penetrating the gear. The impact of the cooling factor within the space suit and of the radiation exchange between aluminum-covered mylar layers was omitted.

The adopted model has the shape of a multilayer cylindrical partition, with a 40 mm inner radius from the axis, which corresponds to the astronaut’s forearm. The remaining parts of the suit may contain electrical elements and other systems, causing changes in the obtained results [31]. In the case of a repeatable shape and thermal conditions, the typical procedure is to introduce only a part of the structure during the calculations. The human forearm implies the cylindrical shape of the garment, i.e., the axisymmetric problem. Therefore, only a part of the axisymmetric structure is introduced, which is a fragment in the form of a circular sector. In order to lessen the computational requirements in the model, only the part with an opening angle of 10° was considered. The distribution of the material layers, thickness and thermal conductivity coefficients were acquired from an available material database [32], as shown in Table 1. The multilayer mylar was classified as a single layer. 

Two extreme cases of extravehicular activity were taken into consideration: a case with the highest susceptibility to the sun’s radiation effect (Hot Case T_a_ = 127 °C) and a case where the sun’s impact was the lowest (Cold Case, T_a_ = −156 °C). Such temperatures are described by NASA as present in the area near the Moon [6]. The CT was set to 37 °C. Heat exchange occurs between the human body and the environment, which makes it possible to insulate the side surfaces of the model. 

The models analyzed here are approximated with the same FEM mesh. The global size of a finite factor was chosen according to the thickness of the thinnest layer (of skin), which is equal to 0.1 mm. A decrease in size below this value would not affect the solution. The computational requirements of the model increase significantly with the number of elements. However, the changes in the accuracy of the obtained results are not visible above the determined mesh density.

A good method to improve the quality of the computational results is to increase the finite element order. Thus, the computational model requires additional nodes at the finite element boundaries, which significantly increases the number of degrees of freedom. The interpolation between the finite elements is consequently nonlinear. The disadvantage of this method is the high computational effort of nonlinear approximations compared with other solutions. The finite element order was set to 1; hence, the approximations of nodal values are described with linear equations [33]. The increase in the finite element order did not cause a notable improvement in the results with a fixed size of the particular element [34,35].

Due to the model’s three-dimensional nature, the *Sweep* function was utilized to permit the transposition of the mesh created on one of the model’s surfaces into the third dimension. Hence, the mesh is even, preventing approximation errors and facilitating the use of other functions.

To solve a case, it is best to utilize the hexagonal elements. Nevertheless, during mesh generation, unwanted tetragonal elements may occur. The application of the *MultiZone* function prevents the creation of tetragonal elements. 

A sudden change in the thermal conductivity coefficient takes place at the meeting point of the materials. In such areas, computational requirements are the highest due to the sudden changes in the parameters. For layers exemplifying the biggest differences in the thermal conductivity coefficient, the *Inflation* function was used, which creates inflation layers with a relatively high value of the proportionality coefficient. The initial thickness of the layer at the edge was 20 μm, with each following generated layer being thicker by 10%. The number of the generated layers is between 3 and 8, depending on the changes in the conductivity coefficient on the layers’ boundaries and on their thickness (Figure 2).

The verification model describes only the heat conduction between the layers of the material. In this way, the distribution of heat areas and the heat flux density are substantially different from the actual model, though still allowing for the assessment of the applied mesh. Only temperatures on the external surfaces of the spacesuit and the astronaut’s body are considered.

The actual model takes into account the heat radiation emission from the surface of the space suit. The suits are covered with white Krylon paint on their surface, which has an emissivity coefficient of ε = 0.98, reflecting a major part of cosmic radiation [36]. The temperatures in the Hot and Cold Cases refer to the ambient temperature. The boundary conditions are summarized for each case in Table 2.

### 2.3. Verification Calculations

The verification calculations were performed for a homogenic material of the resultant parameters. The resultant thermal conductivity coefficient is determined with the transformation of (3).
(6)$$\lambda_{w}=\frac{ln\frac{r_{n}}{r_{0}}}{\sum_{i=1}^{n}\left(\frac{1}{\lambda_{i}}\cdot ln\frac{r_{i}}{r_{i-1}}\right)}$$

Hence, the value λ_w_ = 0.137125. The simplified equation describing the heat flux density on the outer layer of the suit can be defined according to (3), whereby r_0_ and r are radii measured from the axis to the inner and outer layers of the space suit.
(7)$$q=\frac{\lambda_{w}\cdot \left(T_{0}-T_{9}\right)}{r_{9}\cdot ln\frac{r_{9}}{r_{0}}}$$

As a result of the calculations, the following values were obtained: Hot Case, T_9_ = 400.15 K: q = −2234.02 W/m^2^. Cold Case, T_9_ = 117.15 K; q = 4790.74 W/m^2^.

Using Equations (3) and (7), the temperature was determined for any contact surface.
(8)$$T_{i}=T_{i-1}-\frac{r_{9}\cdot q\cdot ln\frac{r_{i}}{r_{i-1}}}{\lambda_{i}}$$

Because radiation is the only form of heat exchange in outer space, the equation of state on the outer layer of the space suit has a following formula:(9)$$\varepsilon \cdot \sigma \cdot \left({T_{9}}^{4}-{T_{\infty }}^{4}\right)=\frac{\lambda_{w}\cdot \left(T_{0}-T_{9}\right)}{r_{9}\cdot ln\frac{r_{9}}{r_{0}}}$$

The outer layer emissivity is equal to ε = 0.98, and the reference emissivity was set to one, similar to that for a black body [24,36]. Therefore, the equivalent emissivity coefficient equal to ε = 0.98 was obtained.

To determine the surface temperature of the suit, Equation (9) was transformed to a fourth-degree polynomial
(10)$${T_{9}}^{4}+{\frac{\lambda_{w}}{{\varepsilon \cdot \sigma \cdot r}_{9}\cdot ln\frac{r_{9}}{r_{0}}}\cdot T}_{9}-T_{0}\cdot \frac{\lambda_{w}}{{\varepsilon \cdot \sigma \cdot r}_{9}\cdot ln\frac{r_{9}}{r_{0}}}-{T_{\infty }}^{4}=0$$

Hence, obtaining polynomial Equation (11) for the Cold Case and (12) for the Hot Case.
(11)$${T_{9}}^{4}+{893383151.197\cdot T}_{9}-277136165592=0$$
(12)$${T_{9}}^{4}+{893383151.197\cdot T}_{9}-302548776871=0$$

By solving the equations, the following temperatures were determined: T_9,C_ = 301.02, K = 27.87 °C; T_9,H_ = 326.01, K = 52.86 °C. 

The density of the heat flux at any point of the model may be determined with the knowledge that the heat flux has a constant value at all points [37]. Thus, after calculating the heat flux density on the outer layer of the space suit, the following equation was established:(13)$$q\cdot 2\pi \cdot r\cdot l=q_{x}\cdot 2\pi \cdot r_{x}\cdot l$$

The heat flux density is hence proportional to the magnitude of the instantaneous radius r_x_.
(14)$$q_{x}=q\cdot \frac{r}{r_{x}}$$

## 3. The Results of the Numerical Calculations

The results encompass the temperature distributions and the heat flux densities for both models. The requirements for the calculation of the mesh are not high when determining the temperature distribution, but during the determination of the heat flux density, the mesh quality plays a key role. In the case of the verification model, the distribution of both parameters for the Hot Case is provided in Figure 3, and for the Cold Case, the distribution is provided in Figure 4.

The temperatures T_sol_ on the boundary of each layer for both models were recorded with the use of the *Probe* function available in Ansys Workbench software. These temperatures were compared with the temperatures T_cal_ calculated in relation to (8) and are summarized in Table 3 for the Hot Case and in Table 4 for the Cold Case; ΔT denotes the relative error. 

A higher relative error is to be observed on the meeting point of the surfaces located furthest from the boundary conditions, i.e., in the center of the material. This stems from the interpolation error of the equations imposed by the Ansys Workbench v. 2023 software. Further nodal values calculated with this method make use of the previously approximated values for the previous nodes, which causes the accumulation of the relative error. The error is also higher for the extreme cold case, where the temperature range is substantially higher than for the extreme heat case. Still, even the biggest relative error in the model, equal to 0.374%, is statistically insignificant. The available literature did not provide any information on the distribution of the temperature and the heat flux density in the space suit.

The verification model (Figure 3 and Figure 4) is determined by the uniform distributions of the temperature fields and the heat flux densities. It follows that the applied computational mesh is sufficiently accurate for the considered problem. The heat flux density distributions (Figure 3b and Figure 4b) have different vector directions for the Hot Case and the Cold Case. For the Hot Case, the heat flux density has a negative value, i.e., the vector is directed from the outside of the model to the inside. On the contrary, the Cold Case is characterized by positive values of the heat flux density. The heat flux densities on the outer surface of the model are equal to q = −2251 W/m^2^ for the Hot Case and q = 4827.2 W/m^2^ for the Cold Case. The accurate values according to Equation (7) are Hot Case: q = −2234.02 W/m^2^ and Cold Case: q = 4790.74 W/m^2^. Thus, the errors are less than 1%. 

Taking into consideration the results and the analysis of the error, it can be said that the results obtained with the proper simulation Ansys Workbench v. 2023 software are reliable. 

In the actual model, the heat emission on the outer layer of the space suit was assumed according to the ambient environment temperature. As a result, the heat distribution and the heat flux density were obtained, as presented in Figure 5 and Figure 6**.** Significantly lower heat flux density values were obtained in comparison with the verification model.

The temperature distribution at the boundary of the actual model layers captured with the *Probe* function is presented in Figure 7. The change in temperature relative to the thickness of around 4.2 mm is linear. This results from the set values of the thermal conductivity coefficient and the small thickness of each layer. The most notable temperature change occurs in the Kevlar layer, which has a significantly smaller thermal conductivity coefficient equal to 0.04 W/mK. It is not possible to utilize it as the main thermal-insulating material because its mass is too big compared with the other textile materials, because of its stiffness and because of the high price. Mylar, which constitutes the main thermal-insulating layer, does not exhibit better thermal-insulating properties in comparison with the remaining materials in the TMG. The most important function of the Kevlar layer should be to limit the emission of heat between each aluminum-covered mylar layer with a low emissivity coefficient [9].

## 4. Discussion

Table 5 contains the results of key parameters for the extreme cases: temperature of the space suit surface T_surf_; skin surface temperature T_skin_; heat flux density on the model surface q_surf_ and on the skin surface q_skin_; and Q. Q, which is a convective heat flux calculated based on the known heat flux density q_skin_ and total surface area of the astronaut’s body. The relationship (2) was used for L = 1.75 m and M_g_ = 75 kg, thus resulting in an S equal to 1.91 m^2^. 

The skin surface temperatures obtained as a result of the modeling are equal to 37.43 °C (Hot Case) and 36.76 °C (Cold Case). In terms of physiological comfort, these values are too high [22]. The obtained heat fluxes for the extreme cases should be interpreted as the astronaut’s metabolic rates sustaining CT = 37 °C. Considering that CT in different body parts has different values [38] and that human body temperature can be maintained in specific boundaries, the heat fluxes in the analysis may be overstated in relation to the actual case. Furthermore, a space suit consists of a structure of many electronic components, which influence the local changes in the process of heat exchange in the outer space environment. Hence, the result from the simulation may be classified as the most dangerous case, as the forearm is usually not covered with any additional element except for the suit’s layup [8].

Conditions imposed by NASA on astronauts during extravehicular activities determine a 15-minute-long exposure to the extreme conditions in space, at a maximal metabolic rate of about 600 W. The metabolic rate achieved in the simulation was decidedly too high in the Hot Case, being equal to 852.36 W. The causes of this can be found in the aforementioned mechanisms present in the actual space suit, for example, cooling the astronaut’s body in extreme conditions. On the other hand, for the Cold Case, the result was equal to 480.05W, making it possible to work for approximately an hour, for which the permissible metabolic rate is around 470 W [8].

## 5. Conclusions

Thermal insulating properties were modeled in an exemplary multi-layer layup of a space suit, both its entirety as well as the specific layers, utilizing computer simulation techniques and physics and mathematical models. The problem was determined using the temperature distributions and the heat flux densities. The results obtained in the computer simulation were compared to the results from the mathematical model, which allowed for the quality assessment of the models. 

A model is always a simplified representation of a space suit. A factor that significantly influences the results is the lack of inclusion of the cooling water system in the LCVG system. It is not feasible to include the flow of the liquid in the model since there is no way to gain access to key information regarding the flow (the temperature of the liquid, its velocity, efficiency, etc.). A similar issue is with the inclusion of the effect of gas enclosed in the pressure garment layer on the heat exchange.

In the future, the impact of liquid cooling and heat emissions between the aluminum-covered mylar layers with a low emissivity coefficient should be considered [9]. This problem can be identified using the Microtomography and the Thermography [39] or statistical analysis [40,41]. Another possibility is the implementation of other mechanisms supporting the thermal insulation of the structure, for example, Phase Change Materials (PCMs) characterized by a high heat capacity, notwithstanding their time-limited effect [38]. However, the analysis of such a solution may be troublesome due to the issues with determining the temperature caused by the difficulties in establishing the variable phase material mass. A further suggested direction is to implement the transient heat exchange state [32,42]. Since one of the mechanisms determining the metabolic rate is sweating, the application of moisture-absorbing materials is worth analyzing further [43]. However, similar to the PCM, it is not a solution that will last for longer periods of time. An improvement in the mesh quality and a more precise approximation of the FEM elements should be deliberate as well [44,45]. The determination of the temperature inside of a space suit may serve as a basis for an optimization of thermal-insulating properties, with the limitation of layer thickness, costs and possible redefinitions of their functions [46,47]. 

## Figures and Tables

**Figure 1 materials-17-00648-f001:**
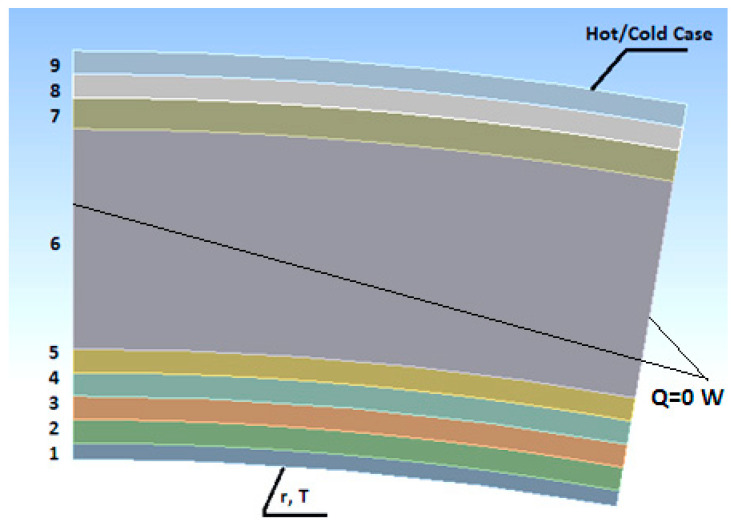
Configuration of the material layers, numeration according to Table 1.

**Figure 2 materials-17-00648-f002:**
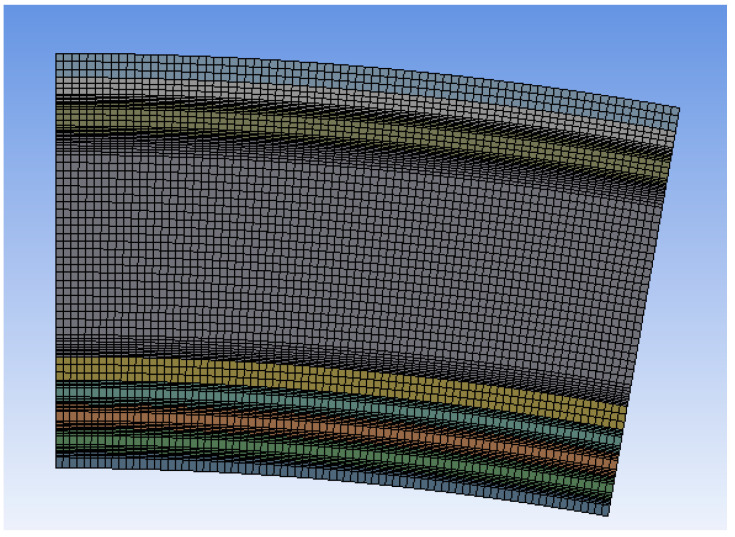
Final version of the finite element mesh.

**Figure 3 materials-17-00648-f003:**
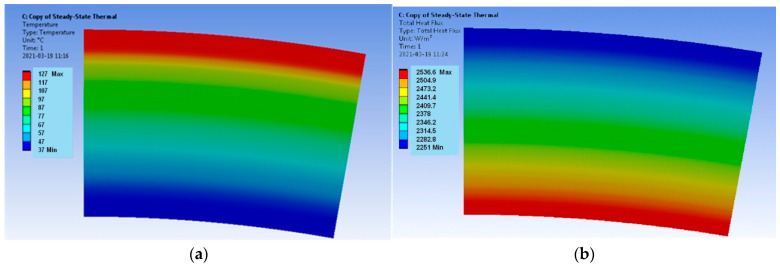
Verification model; Hot Case: (**a**) temperature distribution and (**b**) heat flux density distribution.

**Figure 4 materials-17-00648-f004:**
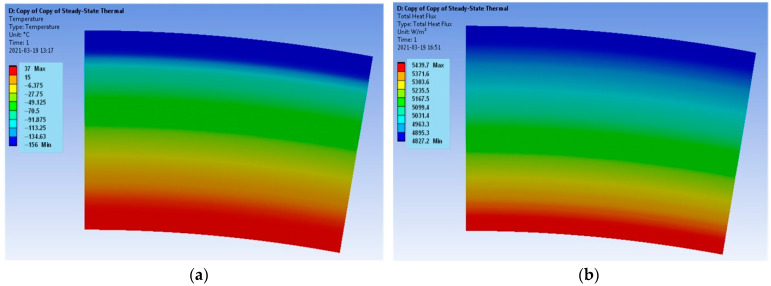
Verification model; Cold Case: (**a**) temperature distribution and (**b**) heat flux density distribution.

**Figure 5 materials-17-00648-f005:**
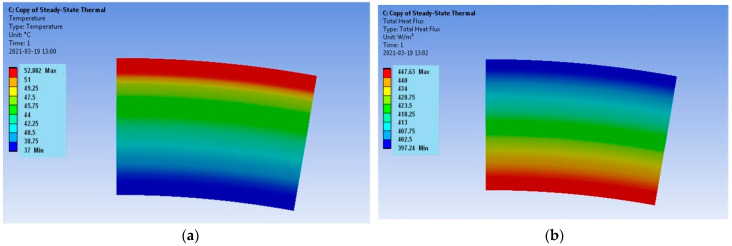
Actual model; Hot Case: (**a**) temperature distribution and (**b**) heat flux density distribution.

**Figure 6 materials-17-00648-f006:**
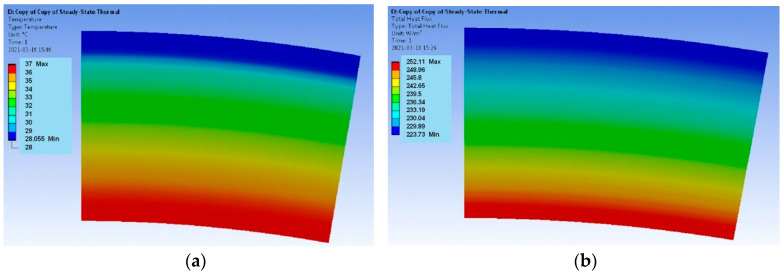
Actual model; Cold Case: (**a**) temperature distribution and (**b**) heat flux density distribution.

**Figure 7 materials-17-00648-f007:**
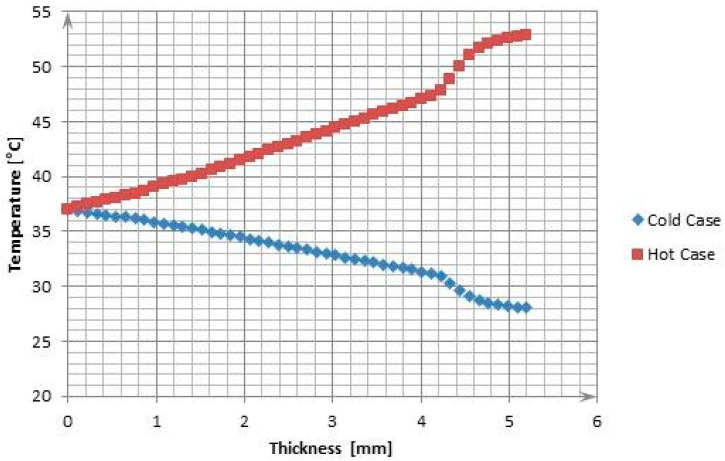
The temperatures at the layer boundary for the outermost elements.

**Table 1 materials-17-00648-t001:** Layer thickness in the model and corresponding thermal conductivity coefficients.

Layer Number	Material	Thickness (mm)	λ (W/mK)
1	Skin	0.2	0.210
2	Nylon	0.3	0.233
3	Nylon	0.3	0.233
4	Dacron	0.3	0.144
5	Nylon	0.3	0.233
6	Mylar	2.8	0.155
7	Kevlar	0.4	0.040
8	Nomex	0.3	0.140
9	Goretex	0.3	0.251
	Total	5.2	

**Table 2 materials-17-00648-t002:** The comparison of the boundary conditions in the models.

Boundary Conditions	Verification Model		Actual Model	
	Hot Case	Cold Case	Hot Case	Cold Case
CT	37 °C			
Outer layer temperature	127 °C	−156 °C	-	
External temperature	-		127 °C	−156 °C
Surface emissivity	-		0.98	0.98
Configuration coefficient	-		0.50	0.50
Insulation on the sides of the model	yes		yes	

**Table 3 materials-17-00648-t003:** The simulation results for the verification model compared to the verification calculations—Hot Case.

Meeting Point of the Materials	T_cal_ (K)	T_sol_ (K)	ΔT (%)
1/2	312.55	312.56	0.005
2/3	315.77	315.71	0.019
3/4	318.97	318.84	0.041
4/5	324.11	323.70	0.124
5/6	327.26	326.78	0.145
6/7	369.89	369.70	0.053
7/8	392.64	392.59	0.012
8/9	397.47	397.45	0.005

**Table 4 materials-17-00648-t004:** The simulation results for the verification model compared to the verification calculations—Cold Case.

Meeting Point of the Materials	T_cal_ (K)	T_sol_ (K)	ΔT (%)
1/2	305.01	304.98	0.011
2/3	298.10	298.23	0.043
3/4	291.24	291.52	0.098
4/5	280.22	281.09	0.310
5/6	273.46	274.49	0.374
6/7	182.04	182.47	0.236
7/8	133.26	133.37	0.079
8/9	122.90	122.93	0.028

**Table 5 materials-17-00648-t005:** The results of the key parameters for the extreme cases.

Key Parameter	Hot Case	Cold Case
T_surf_	326.03 K	301.21 K
q_surf_	−397.24 W/m^2^	223.73 W/m^2^
T_skin_	310.58 K	309.91 K
q_skin_	−446.26 W/m^2^	251.34 W/m^2^
Q	−852.36 W	480.05 W

## Data Availability

Data are contained within the article.

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
