# Peer review of "Numerical Modeling of the Thermal Insulating Properties of Space Suits"

_materials, 2024, doi:10.3390/ma17030648_

Round 1

Reviewer 1 Report

Comments and Suggestions for Authors

In the following, there is a list of questions that the authors should answer before acceptance.

(1)   Line 105-107, the authors say: “The structure consists of 14 layers with the following functions, starting from the human body [9-12], Figure 1.”. Why only 9 layers are shown in Fig.1?

(2)   Line 105-107, the authors say: “The structure consists of 14 layers with the following functions, starting from the human body [9-12], Figure 1.”. But line 108-123, there are a total of 15 layers. Please review and correct the corresponding contents.

(3)   Line 157-158, the authors say: “The perceived thermal comfort is determined by CT=(37±0,3)°C and skin temperature of about 33°C [20].”. It will be better to use the international standard of decimal point symbol.

(4)   The order of tables and pictures are confusing.

(5)   Line 209-210, the authors say: “In order to lessen the computational requirements in the model, only the part with an opening angle of 10° was considered, Figure 1.”. What is the basis for this conclusion? It should be explained in detail.

(6)   Line 220, 'FEM' appeared at the first time, so it should be a full name.

(7)   Line 267 and 272, formula 10 is repeated.

(8)   The quality of Figures 3-6 is poor.

Author Response

The detailed answers are in file Answer Rev 1

Reviewer 2 Report

Comments and Suggestions for Authors

This manuscript presented a numerical modelling study about the thermal insulating properties of space suits. This work has some potential. However, several parts listed below need to be clearer and improved.

Abstract: please clearer the methodology used in this study. In addition, add some numerical results to the abstract.

One general comment: please use “.” (dot) instead “,” (comma) to express all the numerical results presented in this manuscript.

Section 2: I suggest better explore the previously references (Li [14], Li, Zhu [15], Wang et. 129 al. [16]) in Section 2.

Section 2: clearer the novelty of this work.

Table 1: the boundary condition described in Table 1 are based in previously literature?  

Section 4: I suggest better describe the steps and conditions used for obtained the mesh presented in Figure 2. Also add the software used to perform the simulations.

Figure 3 and Figure 4: please better discuss the results presented in both figures.

Page 9 Lines 307-309: the authors said that: “The error is also higher for the 307 extreme cold case, where the temperature range is substantially higher than for the 308 extreme heat case.” Why the error is higher for cold case?

Page 9 Lines 311-317: I suggest compare the results obtained in this work with others from the literature.

Figure 7: please check the names in both axes about correct English spelling.

Table 5: better discuss the results presented in Table 5. In addition, I suggest compare these results with others from the literature.

Conclusion section: several parts in the conclusion section presented a discussion of the obtained results and must be added to the discussion section. I addition, this section is too long and must to be reduced.

Author Response

The detailed answers are in file Answer Rev 2

Reviewer 3 Report

Comments and Suggestions for Authors

The purpose " Numerical modeling of thermal insulating properties of space suits" is to model the thermal insulating properties in an exemplary multi-layer layup of space suits, both its entirety, as well as, the specific layers by utilizing computer simulation techniques and physics and mathematical models. Results obtained in the computer simulation were compared to results from the mathematical model, which allows for the quality assessment of the models."

The topic of the paper is very interesting and useful. What is missing, however, is a qualitative evaluation of the created model.

As for the formal arrangement, a few recommendations

- Edit the paper's layout based on the template - Introduction, Materials and Methods, Results, etc.

- "Nomenclature" is not part of the template for the given journal. I would recommend incorporating individual abbreviations into the text

- In Fig.1 delete the background of the picture - increase the quality

- Use bullet points according to the template - page 3, line 98, p.4 line 139, p. 5, r. 182 etc.

- Line 103 (page 3) - in the description of Figure 1- there is a reference to "Table 2", which, however, is found on page 6, the same with a reference to "Table 1". Edit according to the template "Tables should be placed in the main text near to the first time they are cited".

- Table 1 - what does "tak" mean in the last line? and Table 5 what does "Dane" mean

- In the sentence on line 209- "In order to lessen the computational requirements in the model, only the part with an opening angle of 10° was considered, Figure 1" they refer to the angle in Fig. 1, which, however, is not shown in figure

- use links to figures according to the template for example - (Fig.1)

- Page.11,  Fig. 7 - correct the description of individual axes

- Edit the literature according to the template

Comments on the Quality of English Language

 Minor editing of the English language required

Author Response

The detailed answers are in file Answer Rev 3

Round 2

Reviewer 2 Report

Comments and Suggestions for Authors

After corrections the manuscript reads well. I suggest publication in its current form.

Reviewer 3 Report

Comments and Suggestions for Authors

Accepted in present form

Comments on the Quality of English Language

 Minor editing of the English language required